# (Non-)disclosure of lifetime sexual violence in maternity care: Disclosure rate, associated characteristics and reasons for non-disclosure

**Hannah W. de Klerk**[1,2,3,4]*, **Marit S. G. van der Pijl**[1,2,3,4], **Ank de Jonge**[1,2,3,4], **Martine H. Hollander**[5], **Corine J. Verhoeven**[1,2,3,4,6,7], **Elsa Montgomery**[8], **Janneke T. Gitsels-van der Wal**[1,2,3,4]

1 Amsterdam UMC Location Vrije Universiteit Amsterdam, Midwifery Science, Amsterdam, The Netherlands, 2 Midwifery Academy Amsterdam Groningen, Inholland, Amsterdam, The Netherlands, 3 Amsterdam Public Health Research Institute, Quality of Care, Amsterdam, The Netherlands, 4 Department of General Practice & Elderly Care Medicine, University of Groningen, University Medical Center Groningen, Groningen, The Netherlands, 5 Department of Obstetrics, Amalia Children's Hospital, Radboud University Medical Center, Nijmegen, The Netherlands, 6 Division of Midwifery, School of Health Sciences, University of Nottingham, Nottingham, United Kingdom, 7 Department of Obstetrics and Gynaecology, Maxima Medical Centre, Veldhoven, The Netherlands, 8 Florence Nightingale Faculty of Nursing, Division of Methodologies, Midwifery & Palliative Care, King's College London, London, United Kingdom

* h.deklerk@amsterdamumc.nl

## Abstract

### Background

In maternity care, disclosure of a past sexual violence (SV) experience can be helpful to clients to discuss specific intimate care needs. Little evidence is available about the disclosure rates of SV within maternity care and reasons for non-disclosure.

### Aim

The aim of this study was to examine (1) the disclosure rate of SV in maternity care, (2) characteristics associated with disclosure of SV and (3) reasons for non-disclosure.

### Methods

We conducted a descriptive mixed method study in the Netherlands. Data was collected through a cross-sectional online questionnaire with both multiple choice and open-ended items. We performed binary logistic regression analysis for quantitative data and a reflexive thematic analysis for qualitative data.

### Results

In our sample of 1,120 respondents who reported SV, 51.9% had disclosed this to a maternity care provider. Respondents were less likely to disclose when they received obstetrician-led care for high-risk pregnancy (vs midwife-led care for low-risk pregnancy) and when they had a Surinamese or Antillean ethnic background (vs ethnic Dutch background). Reasons for non-disclosure of SV were captured in three themes: 'My SV narrative has its place

**Data Availability Statement:** The data file is available from the Zenodo database (DOI: 10.5281/zenodo.8195625).

**Funding:** The author(s) received no specific funding for this work.

**Competing interests:** The authors have declared that no competing interests exist.

outside of my pregnancy', 'I will keep my SV narrative safe inside myself', and 'my caregiver needs to create the right environment for my SV narrative to be told'.

## Conclusions

The high level of SV disclosure is likely due to the Dutch universal screening policy. However, some respondents did not disclose because of unsafe care conditions such as the presence of a third person and concerns about confidentiality. We also found that many respondents made a positive autonomous choice for non-disclosure of SV. Disclosure should therefore not be a goal in itself, but caregivers should facilitate an inviting environment where clients feel safe to disclose an SV experience if they feel it is relevant for them.

## Introduction

Sexual violence (SV) is widespread worldwide with reported prevalence rates for women up to 59%, but varying greatly, depending on region, definition, and methodology [1]. SV experiences can affect both physical and mental health in the short and long term and are therefore considered to be a major public health concern [2]. In the Netherlands, a large population study showed that 22% of women reported an SV experience in their lifetime, defined as vaginal, oral, anal, or manual sex against their will. Moreover, sexual harassment, including unwanted kissing and touching in a sexual way, was experienced by 53% of women [3].

Disclosure of the SV experience to a trusted person can benefit survivors of SV. Some women have described telling someone about their SV experience as a 'healing' experience. Telling their story can make them feel less alone and help (re)build trusting relationships with others [4]. Health care professionals are the least likely group women disclose an SV experience to, compared to friends, family, counsellors, and acquaintances [5]. Berry and Rutledge identified five facilitators for disclosure to a physician; concern for medical or physical consequences of the SV experience, direct questioning about SV by the professional, a knowledgeable professional with a positive attitude, the SV experience being relevant to the appointment, and a trusting relationship with the physician [6].

While disclosing SV can be helpful to victims, it can also be damaging, depending on the response they receive. Not being believed and victim blaming are frequently reported and are common responses that people face when telling their stories in personal, as well as in healthcare settings [5,7]. In healthcare settings, in addition to being doubted or blamed, other unhelpful factors that have been identified are: a rushed and uncomfortable environment, lack of privacy and confidentiality, male gender of the provider, irrelevance of SV to the appointment, and lack of a trusting relationship with the provider [6,8].

In maternity care, reasons for non-disclosure of SV experience have not been systematically collected. Qualitative research into personal accounts of women with an SV experience however shows that reasons for non-disclosure can be fear of not being believed [9], fear of being judged [10,11], fear of the child being taken away [9,11], not being asked about an SV experience [10] and concerns about the information being included in medical records that would be shared with other care providers [12]. Some women who disclosed their abuse described how this helped them receive the support they needed. Midwives offered them more frequent antenatal visits and more control over physical examinations which resulted in a positive childbirth experience [13]. Disclosure rates of lifetime SV in maternity care have to our knowledge not been investigated yet and literature on disclosure rates in other healthcare settings are

scarce. One Slovenian study found that only 19.6% of women with a known SV experience ever disclosed this to a care provider (physician, nurse, psychologist or social worker) [14]. Differences in disclosure rates based on client characteristics were not explored.

In Dutch prenatal care guidelines it is recommended that midwives and obstetricians offer universal screening for SV history during the booking appointment. The guidelines do not offer specific recommendations about the conditions, method or wording for the screening or pathways of care after a disclosure [15,16]. The midwifery guideline advises extra care with intimate procedures and preparing the women for these procedures, and referral for psychological help if necessary [15]. The obstetric and midwifery guidelines are taught in graduate education. It is unknown whether maternity care providers adhere to the screening guidelines and what proportion of women disclose an experience of SV to their maternity care provider in the Netherlands.

Understanding disclosure rates within maternity care and possible barriers for disclosure can aid maternity care professionals in adapting to the care needs of their clients. The current study aims to contribute to the understanding of SV disclosure within maternity care by answering the following research questions: (1) what proportion of pregnant women with an experience of SV disclose this to their maternity care provider? (2) Is there a difference in characteristics of pregnant women who had an experience of SV and disclosed this to their maternity care provider compared to women who did not? And (3) what reasons do women have for non-disclosure of SV within maternity care?

## Methods

We conducted a descriptive mixed method study. Data was collected through a cross-sectional online questionnaire with both multiple choice and open-ended items. Mixed methods analysis was performed to answer the research questions. Disclosure rate and associated characteristics were analysed quantitatively and reasons for non-disclosure were analysed qualitatively.

### Ethics statement

On April 14th 2020, the Medical Ethics Committee of the Vrije Universiteit Medical Centre Amsterdam (FWA00017598) stated that the Medical Research Involving Human Subjects Act (WMO) did not apply to our study (2020.084) and the study was exempt from further approval by the committee. Respondents were informed about the research on the questionnaire website, after which they could start the questionnaire. They were advised that filling out the questionnaire constituted their consent to participate in the study. They were able to stop the questionnaire at any time, and answers were not linked to any personal information such as IP address or email address.

### Data collection

Questions about SV were included in a survey study on respectful maternity care in the Netherlands [17]. The digital online questionnaire was a mixture of multiple choice and open-ended questions and was conducted between October 26th and December 17th, 2020. Two client representatives from the Dutch client organisation Birth Movement were involved in developing and piloting the questionnaire. Respondents were reached through social media platforms (Facebook, Twitter and Instagram) and maternity care organisations throughout the country. Organisations, such as midwifery practices and client groups, and social media influencers were approached to help disseminate the questionnaire. Organisations shared the questionnaire on their social media platforms, in newsletters, during prenatal educational evenings and on posters. Social media influencers shared a post or a story about the questionnaire on

Instagram. Respondents were offered the chance to win a gift card worth 50 euros after completing the survey.

To be eligible to participate in the study, respondents had to be 16 years of age or older and their last birth had to be no more than 5 years ago at the time of filling out the questionnaire (between 2015 and 2020).

## Measurements

Three questions about SV were included in the questionnaire. First, all respondents were asked whether they had ever experienced SV (answer categories: yes/no/prefer not to answer). Lifetime SV was defined for the respondents as 'undergoing or performing sexual things you did not want to do'. This definition is in line with a Dutch sexual health population study for comparison (3). Second, to measure the disclosure rate of SV experiences in maternity care, respondents who reported SV in the questionnaire were asked whether they had disclosed the SV experience to their maternity care providers (answer categories: yes/no/prefer not to answer). Last, the respondents who indicated they had *not* disclosed SV to their maternity care providers were asked to elaborate on the reason for non-disclosure in an open text field.

Background characteristics of the respondents that were included in the current study were age, education, ethnicity and relationship status. All characteristics were self-reported by the respondents. Age, education at the time of the last birth and ethnicity were grouped in similar categories as the Dutch sexual health population study [3] for comparison. Age at the time of the last birth was measured in years and grouped in 'younger than 25 years', '25 through 35 years' and '35 years or older'. Respondents who had not finished any education, or whose highest completed education was primary school or preparatory vocational education were grouped in 'basic / no education'. Respondents who had completed high school or a vocational education were grouped in 'intermediate education' and respondents who completed at least a bachelor's degree were grouped in 'advanced education'. Respondents who were born in the Netherlands *and* whose parents were both born in the Netherlands were assigned the 'Dutch' ethnicity. Respondents born in another Western country (as defined by the Dutch National Statistics Office [18]) *or* with at least one parent born in another Western country were assigned a 'Western' ethnicity. Dutch and Western were grouped together as 'Dutch/Western ethnicity'. The largest immigrant groups in The Netherlands are people with a Moroccan, Turkish, Surinamese, and Antillean background respectively. Respondents who were born in Morocco or Turkey *or* whose parents (at least one) were born in Morocco or Turkey were assigned to the group Moroccan/Turkish ethnicity. Similarly, people from or with parents from Surinam or the Antilles were assigned to the group Surinamese/Antillean ethnicity and respondents from or with parents from other non-Western countries were assigned to the group 'other non-Western' ethnicity. Consequently, Moroccan/Turkish, Surinamese/Antillean and other non-Western ethnicities in our study include first and second generation immigrants, but not third generation immigrants. The proportion of non-Western third generation immigrants in the reproductive phase in the Netherlands is estimated to be about 1% [19]. Relationship status was defined as either single or in a relationship (married or otherwise).

Obstetric characteristics used in this study were parity and care provider at the start of pregnancy. Respondents were grouped in either 'primiparous' or 'multiparous'. In the Netherlands, depending on the perceived level of risk in pregnancy, clients enter maternity care and have their initial booking appointment either through a community-based midwifery practice (low risk) or through the hospital (high risk). In some regions all women have the initial booking appointment at a community-based midwifery practice and are subsequently referred to hospital when considered high risk. Respondents were grouped as having received either

'midwife-led care' or 'obstetrician-led care' at the start of pregnancy. This indicates where the booking appointment took place and thus who performed SV screening as part of history taking.

## Quantitative analysis

The frequency and percentage of women who reported an SV experience and disclosed this to their maternity care provider were calculated and reported. Background and obstetric characteristics were calculated and frequencies and percentages were reported for the study sample of women with a reported SV experience.

Binary logistic regression analysis was performed to investigate what background and obstetric characteristics were associated with disclosure of SV to a maternity care provider. First, simple binary regression analyses were done for all characteristics separately, and second, a multiple logistic regression analysis was performed. The multiple regression model included all background and obstetric characteristics, to explore associations between respondents' characteristics and disclosure of SV while controlling for confounding effects between the characteristics. The sample size of >1000 respondents was sufficiently large for the six characteristics to be included. Odds ratios of disclosure of SV, confidence intervals and p-values were reported. P-values <0,05 were considered statistically significant. The quantitative analyses were performed in Statistical Package for Social Sciences (SPSS) version 26.0 for Windows (SPSS Inc., Chicago, IL, USA).

## Qualitative analysis

To answer the third research question (what reasons do women have for non-disclosure of SV), we performed a thematic analysis. We followed the six steps of reflexive thematic analysis as developed by Braun and Clarke; (1) familiarisation with the dataset, (2) coding, (3) generating initial themes, (4) developing and reviewing themes, (5) refining, defining and naming themes and, (6) writing up [20,21].

The reflexive thematic analysis allowed us a primarily inductive analysis of a relatively large dataset of over 400 answers to the open-ended survey question. We took a critical realist orientation to our analysis, meaning that we approached the data not as a reflection of an independent reality to be discovered, but as a reflection of the respondents' particular representation of reality. This orientation allowed both literal interpretations of the women's answers and at the same time development of a more theoretical understanding of central concepts around decision-making surrounding SV disclosure within the social context of Dutch maternity care. For example, one respondent answered "it was only once" to the question why she had not disclosed her SV experience. To this respondent, this 'observable' fact evidently led to non-disclosure. A single SV event is, however, not taken literally as a reason not to disclose, because for others a single SV event could be highly relevant to the pregnancy. The answer was understood as the respondent explaining why she felt her SV experience was not severe and therefore did not bear relevance for her in the context of maternity care. Furthermore, explaining why the experience was not so bad, was understood as a particular way to construct the SV narrative so that it was a chapter outside of the pregnancy.

After familiarisation with the dataset, HK performed the initial inductive coding and generated initial themes. This first stage of coding and theme development was primarily inductive and focused on semantic content. Themes were further discussed, developed and reviewed against the codes between HK, MP and JG. A narrative psychology framework was chosen at this stage to further develop the themes. The themes were further refined, named and agreed

upon by all authors. HK finished the analysis by writing up the results. We used MAXQDA 2020 (VERBI Software, 2019) for qualitative data analysis.

## Results

### Study sample

A total of 12,240 respondents filled out the online questionnaire (Fig 1), 1,219 of whom omitted the SV prevalence question and 148 indicated they preferred not to answer. These respondents also did not answer most of the other personal background items leading up to the SV question, indicating they did not wish to disclose personal information in general, not SV history specifically. Of the 10,873 respondents who answered the SV question, 1,126 respondents (9,6%) reported an SV experience in the questionnaire. Five of these respondents omitted the SV disclosure question and were therefore excluded from further analysis. Also, one respondent was excluded because she did not receive any maternity care during her pregnancy. A total of 1,120 respondents were left in the study sample.

Most respondents were between 25 and 34 years of age (72,5%), had a Dutch or other Western ethnicity (92.8%), completed advanced education (vocational or university, 63,5%) and were in a relationship (97,2%). Furthermore, over half of the respondents had given birth to their first child (58,3%) and most entered maternity care through a community-based midwifery practice (81,9%) (Table 1).

### Disclosure rate

Of the 1,120 respondents with a reported SV experience, 581 (51.9%) had disclosed this experience to their maternity care provider and 539 (48.1%) had not (Fig 1).

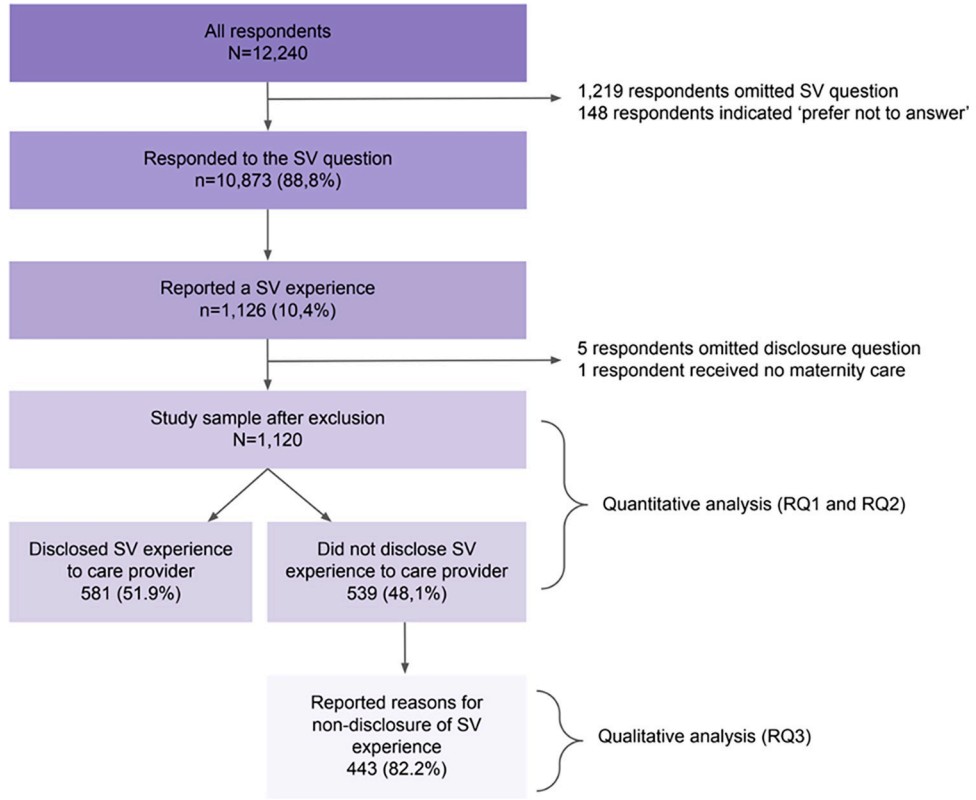

**Fig 1. Overview of selection of study sample.**

**Table 1. Characteristics of respondents in study sample (N = 1,120).**

| Characteristic | Respondents n (%)* |
|---|---|
| **Age** | |
| **<25 years** | 93 (8.3) |
| **25–34 years** | 810 (72.5) |
| **>34 years** | 214 (19.2) |
| **Ethnicity** | |
| **Dutch/Other Western** | 1,040 (92.8) |
| **Moroccan/Turkish** | 11 (1.0) |
| **Surinamese/Antillean** | 31 (2.8) |
| **Other non-Western** | 38 (3.4) |
| **Education** | |
| **Basic / no education** | 24 (2.2) |
| **Intermediate** | 370 (33.3) |
| **Advanced** | 716 (63.5) |
| **Relationship status** | |
| **With partner** | 1,088 (97.2) |
| **Single** | 31 (2.8) |
| **Parity** | |
| **Primiparous** | 653 (58.3) |
| **Multiparous** | 467 (41.7) |
| **Care model at start of pregnancy** | |
| **Midwife-led care** | 917 (81.9) |
| **Obstetrician-led care** | 203 (18.1) |

*Total of respondents varies per characteristic due to some missing answers.

## Associations with non-disclosure

Associations between background and obstetric characteristics and disclosure of SV are presented in Table 2. Respondents who received obstetrician-led care at the start of pregnancy were less likely to disclose an SV experience to their maternity care provider than women who received midwifery-led care (OR 0.58, 95% CI: 0.42–0.80). Also, Surinamese and Antillean women were less likely to disclose SV than Dutch and other Western women (OR 0.46, 95% CI: 0.21–0.99). The multiple regression model explained only a small proportion (Nagelkerke pseudo $R2 = 0.036$) of the variance in disclosure of SV.

## Reasons for non-disclosure

Of the 581 women who did not disclose their reported SV experience to a maternity care provider, 443 (76.2%) elaborated on their reasons for non-disclosure in the open-ended question (Fig 1). The answers ranged from one word to four sentences long. After data familiarisation, initial coding and theme development, it became apparent that the framework of narrative psychology was helpful in understanding the reasons women do not disclose an SV experience to their maternity care provider. Narrative psychology is founded in the work of philosopher Ricoeur, who theorised that a person's identity is not a stable entity, but the result of the continuous telling and retelling of a person's life story [22]. A narrative here is not just a story, but a way of making sense of the past and future. The SV experience was more than an isolated event for the women in our study, but also a narrative that was related to what the SV experience meant for the pregnancy. The SV narrative was not always congruent with the pregnancy narrative the women wished to tell within the context of maternity care. The three themes we developed are formulated in the language of this theory and are concerned with locating the SV narrative in relation to pregnancy and maternity care, specifically: (1) 'My SV narrative has

**Table 2. Associations between respondent characteristics and disclosure of sexual violence experience (n = 1,120).**

|  | Disclosure of SV n = 581 (%) | Non-disclosure of SV n = 539 (%) | Crude OR (95%CI) | Adjusted OR (95%CI)* |
|---|---|---|---|---|
| **Age** |  |  |  |  |
| <25 years | 54 (58.1) | 39 (41.9) | ref | ref |
| 25–34 years | 424 (52.3) | 386 (47.7) | 0.91 (0.73–1.12) | 0.99 (0.62–1.58) |
| >34 years | 102 (47.7) | 112 (52.3) | 0.88 (0.75–1.03) | 0.87 (0.51–1.49) |
| **Ethnicity** |  |  |  |  |
| Dutch/Other Western | 545 (52.4) | 495 (47.6) | ref | ref |
| Moroccan/Turkish | 4 (36.4) | 7 (63.6) | 0.72 (0.39–1.34) | 0.49 (0.14–1.71) |
| Surinamese/Antillean | 10 (32.3) | 21 (67.7) | **0.76 (0.59–0.98)** | **0.46 (0.21–0.99)** |
| Other non-Western | 22 (57.9) | 16 (42.1) | 1.06 (0.90–1.25) | 1.13 (0.58–2.23) |
| **Education** |  |  |  |  |
| Basic / no education | 17 (70.8) | 7 (29.2) | ref | ref |
| Intermediate | 198 (53.5) | 172 (46.5) | 0.79 (0.55–1.14) | 0.47 (0.19–1.18) |
| Advanced | 361 (50.4) | 355 (49.6) | 0.82 (0.65–1.04) | 0.41 (0.16–1.03) |
| **Relationship status** |  |  |  |  |
| With partner | 561 (51.5) | 528 (48.4) | ref | ref |
| Single | 20 (64.5) | 11 (35.5) | 1.71 (0.81–3.60) | 2.11 (0.95–4.71) |
| **Parity** |  |  |  |  |
| Primiparous | 356 (54.5) | 297 (45.5) | ref | ref |
| Multiparous | 225 (48.2) | 242 (51.8) | **0.78 (0.61–0.98)** | 0.82 (0.64–1.05) |
| **Care model** |  |  |  |  |
| Midwife-led care | 499 (54.4) | 418 (45.6) | ref | ref |
| Obstetrician-led care | 82 (40.4) | 121 (59.6) | **0.57 (0.42–0.77)** | **0.58 (0.42–0.80)** |

*Adjusted for age, ethnicity, education, relationship status, parity and care model at start of pregnancy.

its place outside of my pregnancy', (2) 'I will keep my SV narrative safe inside myself', and (3) 'my caregiver needs to create the right environment for my SV narrative to be told'. We present these themes with illustrative data examples.

## Theme 1: My SV narrative has its place outside of my pregnancy

The most prominent theme was the separation of the SV narrative from the pregnancy, either because women felt the experience simply had no influence on the pregnancy narrative, or because they did not *want* the experience to have any influence over the pregnancy narrative. The majority of the women in our study felt that the SV experience did not affect them in general; "*No influence on my life, my wishes, or my choices*" (R38) and "*I am able to separate these two things and it doesn't have anything to do with the birth for me*" (R58). Women also reflected on the reason *why* the SV experience did not affect their pregnancy narrative. Often the SV experience was believed to be not so bad: "*Because it fell under the incidents that every woman experiences (pushy men)*" (R306); "*It was in my younger years with another young child who forced me. Luckily, it wasn't extreme and did not have an impact as such, causing me any trouble now*" (R74) and "*It was only once*" (R98). Also, when sufficient time had passed since the SV experience, for some women this placed the experience so much in the past that it automatically lost relevance, or time had allowed women to find closure before they got pregnant; "*This happened when I was 7 years old and I've left it far behind me*" (R37),; "*I processed this in therapy and put it to rest*" (R20) and "*I was treated for this with EMDR*" (R276). Additionally, for some women, previous intimate care experiences had proved to be unproblematic, confirming irrelevance of the experience to the pregnancy: "*I also noticed before, that during the IUD placement and during gynaecological exams it didn't have a (negative) effect on me*" (R203) and "*after 3 births I knew it wouldn't bother me the 4th time*" (R123).

While most women simply concluded that the SV experience had no influence on the pregnancy, a few were actively pushing the narrative away from the pregnancy: "*I didn't want to taint the birth with this subject*" (R385) and "*I wanted to go through pregnancy positively and not give it any negative undercurrent*" (R154).

All these women, whether motivated by a sense of genuine closure or by the urge to repress a potentially negative influence, made an active decision not to disclose their SV history and leave the SV narrative outside of the pregnancy. The pregnancy narrative had no place for a negative past experience that was judged to be, most likely, of no consequence. In some cases, however, women reflected on the regrets of a non-disclosure in hindsight:

"*I didn't perceive it as sexual violence yet. I didn't think it was worth mentioning it because I wasn't raped. But the birth made me think about the many situations where I was harassed, pushed into sexual acts against my will. I would want to disclose this to a midwife now because these 'little' things can cause you not to feel in control of your own body. I didn't feel like that at all during my birth.*" (R188)

"*Because I didn't think it was relevant because I'd already had various therapies with a psychologist and sex therapist and left it behind me. I didn't want it to control my life after all those years. During the recovery of my pelvic floor problems after the birth, I realised I should have disclosed this.*" (R348)

## Theme 2: I will keep my SV narrative safe inside myself

This theme reflects the vulnerability of some women who were private about their SV narrative and were often afraid of the judgements and responses of the outside world and maternity care providers in particular. To them, at that point in time, the safest place to keep the SV narrative was inside themselves.

Some women had never talked about the SV experience which was voiced as a self-evident reason not to disclose to the maternity care provider; "*No one knows*" (R98) and "*I didn't tell anyone*" (R138). Other women had not yet processed the SV experience which made the narrative too vulnerable to disclose in the maternity care situation: "*Still processing*" (R259) and "*Because this is a major trauma for which I had been in therapy at the time. This stopped when I got pregnant and was resumed afterwards*" (R267). A sense of shame was sometimes the reason to keep the SV narrative inside: "*I am ashamed of this and prefer not to share it for that reason*" (R152) and "*Because I don't feel comfortable to talk about that out of shame*" (R347). Some women also expressed a fear of bias from caregivers after a disclosure: "*Afraid of the prejudices*" (R4) and "*Didn't want to be treated differently*" (R2).

## Theme 3: My caregiver needs to create the right environment for my SV narrative to be told

This theme encompasses how maternity care professionals failed to provide the right circumstances for an SV narrative to be disclosed. It was not always clear whether different circumstances would have led to a disclosure, but the women viewed a safe and inviting maternity care environment as conditional to the *possibility* of a disclosure.

Many women did not disclose the SV experience because the caregiver had not asked about it: "*I wasn't asked about it, so I didn't know I could*" (R101); "*Wasn't asked about it and I thought it was about my child and not about me. I didn't realise it could be relevant*" (R97) and "*It didn't cross my mind it could be relevant, could have been helpful*" (R231). Simply screening for SV was not always enough, however, to create a safe environment for a disclosure. Some

women felt uncomfortable or even unsafe to disclose the SV narrative to their caregivers, because the screening question was asked in the presence of a third person: "*My partner was with me that time and I felt uncomfortable to indicate this*" (R221); "*My mother happened to be there together with me and my partner during the booking appointment. My mother doesn't know about this.*" (R220) and "*Ex-partner was always present who was the perpetrator himself*" *(R35).*

Sometimes, a trusting relationship with the care provider was seen as a prerequisite for disclosure and lacking as such: "*I find it very difficult to talk about myself with a stranger*" (R153), "*All these different caregivers made it too unsafe to tell. Trusting relationship could not be built*" (R3) and "*At every appointment you see someone else and there is already no room for a personal approac*h" (R335). Based on a different relationship, some chose to disclose to their community midwife, but not to the hospital-based caregivers after a referral: "*Did talk about it with my community midwife. From 27 weeks on I became high risk and was referred to the hospital. Because I didn't have a regular midwife/gynaecologist I didn't feel the need to share something so personal*" (R217) and "*My doula and community midwife did know about it. I didn't feel comfortable with the hospital midwife, she also didn't ask*" (R219).

Lastly, some women expressed concerns about the SV experience written down in the medical file where they might lose control over the narrative: "*I didn't want it in writing in association with my name*" (R435) and "*In my opinion, discussing it (and writing it down in your file) can also sometimes have a stigmatising effect in a way that I don't think contributes to the good support I received from the midwife*" (R301).

## Discussion

Half of the respondents who reported an SV experience in the current study disclosed this experience to their maternity care provider. Women were less likely to disclose when they received obstetrician-led care than women who received midwife-led care at the start of pregnancy and women with a Surinamese or Antillean ethnic background were less likely to disclose than women with an ethnic Dutch background. When deciding not to disclose SV to a maternity care provider, there were three main reasons: (1) 'My SV narrative has its place outside of my pregnancy', (2) 'I will keep my SV narrative safe inside myself', and (3) 'my caregiver needs to create the right environment for my SV narrative to be told'.

The proportion of disclosure found in this study was surprisingly high. The only other study into disclosure rate of SV in healthcare that we know of, which investigated disclosure at any point in time to any health care provider on a maternity ward in Slovenia, reported an overall disclosure rate of 19,6% [14]. Possibly, the Dutch maternity care policy of universal screening at booking appointments invites more disclosures than other healthcare settings. Also, SV is likely to be underreported in our study (as described in more detail in the limitations paragraph): the prevalence rate was 10.2%, compared to 34% in the female Dutch population [23]. It is likely that women who did experience, but did not report an SV experience in the questionnaire, did not report this to their maternity care provider either. Consequently, the disclosure rate in this study may be overestimated.

Midwife-led care, in contrast to obstetrician-led care, was associated with more women disclosing SV. The midwife-led care model with one midwife or a small team of midwives providing prenatal, natal and postnatal care, offers continuity of care more often than an obstetrician-led care team that generally constitutes all care providers from the obstetric unit. In a smaller team, a trusting relationship between the client and the midwife can be built over time. A trusting relationship was found to be a facilitator for disclosure in the current study, which confirms previous studies [6,8]. Indeed, some women in our study chose to disclose to

their community midwife, but not to their hospital-based caregivers subsequently because of the different nature of the relationship. It is also possible that hospital-based maternity care providers are less likely to screen for SV because more than 90% of women in The Netherlands start their care with a community midwife who takes the general history early in pregnancy and women are usually referred to hospital at a later stage.

The other (negative) association with disclosure of SV that we found was a Surinamese or Antillean ethnic background compared to a Dutch ethnic background. An explanation for this finding was not found in the qualitative part of our study, but may be found in intersectional research. Ullman and Filipas found that women from ethnic minority groups (mostly African-American) in the United States report more negative social reactions when they disclose SV than white Americans [24]. Black women in white societies suffer from historically charged, hypersexualized racial stereotypes. They are often portrayed as sexually loose and willing and, therefore, 'cannot' be raped. This delegitimisation of black women as victims leads to being believed less often after disclosure of SV [25,26]. Fear of stigmatisation and not being believed may consequently inhibit an SV disclosure in black women.

Although the care model and Surinamese or Antillean ethnicity were associated with disclosure of SV in this study, it is important to note that our multiple regression model explained only a small proportion of the variance in disclosure rate. In other words, the background and obstetric characteristics we investigated were not the most important reasons why some women did, and others did not disclose. Therefore, the reasons for non-disclosure that women gave in the qualitative part of our study may be more relevant for non-disclosure than the investigated respondent characteristics.

We found that a safe and inviting maternity care environment was conditional to a disclosure of SV, as captured in the care environment theme of our qualitative analysis. A lack of screening, a lack of a trusting relationship and concerns about stigmatisation after the SV narrative was written down in the file were reasons for non-disclosure in our study and were found in previous studies as well [10–12]. Additionally and importantly, we found that when maternity care providers screened for SV, the presence of a third person, most often the partner, was another reason for non-disclosure of SV. Moreover, there were situations where the SV screening happened in the presence of the partner who was also the perpetrator. SV occurring in a current partner relationship is a form of intimate partner violence. The World Health Organization advises privacy and confidentiality when screening for intimate partner violence in pregnancy, to ensure a safe situation [27]. Two important points can be taken from the care environment theme of our analysis. Firstly, the Dutch maternity care guideline recommendations of universal SV screening is not always followed. Secondly, to ensure a safe screening environment, guidelines require more specific recommendations addressing the presence of a third person (privacy), a trusted care provider and client agency over medical records (confidentiality).

An unsafe care environment is a negative reason for non-disclosure of SV. Similarly, negative reasons for non-disclosure of SV can be feelings of shame, fear of judgement or an unprocessed trauma, as captured in our theme 'keeping the narrative safe inside', confirming previous research [10,11]. This study adds to the existing literature by uncovering a more positive reason for non-disclosure of SV. The majority of our respondents simply felt that the SV narrative had no role in their pregnancy because it had not been so bad, it was a long time ago, it was processed properly, or previous care experiences proved that it would not be an issue. The overwhelmingly negative reasons for non-disclosure found in the available qualitative literature is likely due to the selection of respondents. The respondents all participated exactly *because* their SV experience had had a substantial impact on their pregnancy, in contrast to the respondents of our systematically collected sample. We found that non-disclosure of SV experience can be a positive autonomous choice. Therefore, disclosure should not be the goal of

 

screening for SV in itself, but screening should be a signal to the client that if she judges her SV experience relevant to her pregnancy she *can* disclose if she so wishes.

## Strengths, limitations and future directions

This study has several strengths. To our knowledge, it is the first to systematically investigate disclosure rate of SV and associations with respondent characteristics in maternity care and to collect reasons for non-disclosure in a large group of women. Due to the method of data collection through various channels including influencers and maternity care organisations, we had a large and unique sample of over 1,000 women with an SV experience.

The study has some limitations as well. SV was likely underreported in the questionnaire, which presumably led to an overestimation of the disclosure rate of SV to the maternity care provider. There are several possible explanations for underreported SV in our questionnaire. Firstly, we used a one item question about lifetime SV. A large nationwide Dutch sexual health study showed that using a series of specific questions about SV resulted in a higher prevalence rate of SV than a single item question [23]. However, they still found a prevalence of 34% amongst Dutch women with the same single item question that we used in the current study. An explanation for underreporting of SV in our study was that the questionnaire focused on a different subject than SV. The main aim of the questionnaire was to evaluate women's experiences of the care they received during their last birth, with a focus on respectful maternity care. Therefore, respondents may not have been prepared for and motivated to report on this sensitive matter. Furthermore, the SV question was the last item of a questionnaire that took respondents on average 15–30 minutes to complete. This may have also led to little motivation to answer the SV item.

Another limitation of this study is that no service users with lived experience were involved in the design or analysis of the research. Representatives of the Dutch Birth Movement were however involved in the design of the study. Despite efforts to reach respondents from different backgrounds through various channels, the respondents in the overall study sample were older, had more education, were more often ethnic Dutch and were more often pregnant with their first child than the Dutch population [17]. This is reflected in our study sample of 1,120 respondents as well. It is possible that groups that were underrepresented in our study have different reasons for non-disclosure than the reasons we have found in the qualitative analysis. Importantly, we found that respondents with a Surinamese or Antillean background were less likely to disclose SV than women with a Dutch background, and at the same time, this group was underrepresented in our study. We recommend that future research into disclosure of SV experience explicitly take the perspective of women with a non-white background into consideration. Furthermore, as our study focused on non-disclosure, future research should investigate how women who disclosed an SV experience to their maternity care provider evaluate the support they were offered after a disclosure and whether they felt stigmatised or otherwise treated negatively after a disclosure.

## Conclusion

This study shows that in Dutch maternity care, there is a relatively high level of SV disclosure, likely due to a universal screening policy. However, substandard care factors surrounding the screening for sexual violence were also described by the participants; screening was done in the presence of third persons, by caregivers without a trusting relationship with the client and without addressing confidentiality and medical records. We recommend that guidelines that suggest (universal) screening for SV also specify recommendations to create a safe environment for a disclosure of SV.

 

We found that many respondents made a positive autonomous choice for non-disclosure of SV. Disclosure should therefore not be a goal in itself, but caregivers should facilitate an inviting environment where clients feel safe to disclose an SV experience if they feel it may be relevant for their pregnancy and maternity care.

## Acknowledgments

We thank all the women who participated in the questionnaire, and sometimes accessed difficult memories to do so.

## Author Contributions

**Conceptualization:** Hannah W. de Klerk, Marit S. G. van der Pijl, Ank de Jonge, Corine J. Verhoeven, Elsa Montgomery, Janneke T. Gitsels-van der Wal.

**Data curation:** Hannah W. de Klerk, Marit S. G. van der Pijl.

**Formal analysis:** Hannah W. de Klerk.

**Investigation:** Hannah W. de Klerk, Marit S. G. van der Pijl, Ank de Jonge, Martine H. Hollander, Corine J. Verhoeven.

**Methodology:** Hannah W. de Klerk, Marit S. G. van der Pijl, Ank de Jonge, Martine H. Hollander, Corine J. Verhoeven, Elsa Montgomery, Janneke T. Gitsels-van der Wal.

**Project administration:** Hannah W. de Klerk, Marit S. G. van der Pijl.

**Supervision:** Ank de Jonge, Janneke T. Gitsels-van der Wal.

**Validation:** Marit S. G. van der Pijl.

**Writing – original draft:** Hannah W. de Klerk.

**Writing – review & editing:** Hannah W. de Klerk, Marit S. G. van der Pijl, Ank de Jonge, Martine H. Hollander, Corine J. Verhoeven, Elsa Montgomery, Janneke T. Gitsels-van der Wal.

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
