## [Decision Letter · Decision Letter 0]

18 Jul 2023

PONE-D-23-12676(Non-)disclosure of lifetime sexual violence in maternity care: Disclosure rate, associated characteristics and reasons for non-disclosurePLOS ONE

Dear Dr. de Klerk,

Thank you for submitting your manuscript to PLOS ONE. After careful consideration, we feel that it has merit but does not fully meet PLOS ONE’s publication criteria as it currently stands. Therefore, we invite you to submit a revised version of the manuscript that addresses the points raised during the review process.

We look forward to receiving your revised manuscript.

Kind regards,

I. Marion Sumari-de Boer, Ph.D

Academic Editor

PLOS ONE

Journal Requirements:

In addition, you indicated that ethical approval was not necessary for your study and that the Medical Ethics Committee of the VU University Medical Centre Amsterdam (FWA00017598) stated that the Medical Research Involving Human Subjects Act (WMO) did not apply to your study (2020.084) and the study was exempt. Please could you include a copy of the exemption letter from the Medical Ethics Committee as an ""Other"" file.

4. Please amend your manuscript to include your abstract after the title page.

Reviewers' comments:

Reviewer's Responses to Questions

**Comments to the Author**

1. Is the manuscript technically sound, and do the data support the conclusions?

Reviewer #1: Yes

2. Has the statistical analysis been performed appropriately and rigorously? 

Reviewer #1: Yes

3. Have the authors made all data underlying the findings in their manuscript fully available?

Reviewer #1: No

4. Is the manuscript presented in an intelligible fashion and written in standard English?

Reviewer #1: Yes

5. Review Comments to the Author

Reviewer #1: Thank you for the opportunity to review this extremely well written mixed-methods study that contributes to the evidence base informing appropriate maternity care for women with a history of sexual violence. It is approached in a sensitive and engaging way with clear practice and research recommendations. Some minor issues to address are detailed below:

Abstract: The setting is not clear in the title or abstract. It would also be useful to have a very brief overview of maternity care provision in the Netherlands for context here.

Abbreviations should be written in full at first mention.

Introduction:

Line 60: When describing Dutch prenatal care guidelines it would be useful to elaborate on the purpose for universal screening. For example is there adequate support service available if sexual violence is disclosed? Are professionals trained in how to screen for SV? Does this include any kind of trauma informed training?

Methods: Although it is clear that relevant organisations and individuals were approached to disseminate the questionnaire, other public involvement and engagement is not mentioned in the methods. Were any service users with lived experience involved in the design or analysis of the research? If not it would be useful to revisit this in the discussion under 'strengths and limitations' to inform future research in this area.

Line 107: Regarding background characteristics of participants: Using country of birth (or parents country of birth) an interesting measure of ethnicity and I think it would be useful to highlight it's limitations.Fpr example does it account for 2nd/3rd etc generation migrants, or those who were born in the Netherlands who identify as Black? This is important as those who identify as having an ethnic minority background may not be identified using 'country of birth' but still have different experiences of care such as discrimination and stigma based on their ethnicity.

Line 130: Please clarify which maternity setting is the normal pathway for high/low risk women, for example do those women with 'high risk' obstetric or medical factors always experience hospital care as this is the only obstetric led setting?

Line 135: What was the justification to only provide characteristics for those women who disclosed SV experience rather than compare characteristics for those who did and did not disclose? Is this something that might be presented in a further analysis of the data?

Line 157: It is not clear how the critical realist sense was incorporated into the analysis, perhaps an example would be useful here?

Results: A short paragraph describing the characteristics presented in Table 1 is required. Again, is there any scope to compare these with all questionnaire respondents (not necessarily in this paper but in a future analysis to explore characteristics associated with experience of lifetime SV)?

Line 333: I wondered if these quotations were better placed under theme 2 as they seem to better relate to keeping the narrative and perceptions of stigma/fear of disclosure?

Discussion:

Line 359: Does the midwife led care model always offer continuity of care? To what extent? How many women receive continuity of care in the Netherlands? What is the policy background? It seemed to me that a lack of continuity was evident in the qualitative findings or dis this only relate to the hospital/obstetric led setting?

Line 374: There is a substantial evidence base of black women not being listened to or believed that might be a useful contribution to this discussion. It might also be useful ti highlight potential language barriers around disclosure here.

Line 378: The last sentence needs further explanation to make sense.

Line 383: What is it about written information what women might fear?

Line 423: Please expand on this point; what was the main focus of the questionnaire?

Regarding future research recommendations, it would be useful to explore the experiences of those who did disclose SV to a maternity care provider. For example how did they experience access to support services? Were fears of social care involvement relieved/substantiated and how?

6. PLOS authors have the option to publish the peer review history of their article (what does this mean?). If published, this will include your full peer review and any attached files.

Reviewer #1: **Yes: **Dr Hannah Rayment-Jones

---

## [Author Response · Author response to Decision Letter 0]

9 Sep 2023

Thank you for providing important feedback that helped us improve our paper. We have answered all the points raised and made changes to the manuscript accordingly. Below you will find our point-by-point reply to all comments in italics. Changes in the revised manuscript are made with track changes; both a version with track changes and a clean version are uploaded.

We followed the PLOS ONE style requirements for our revision. 

We elaborated on participant consent in the Ethics statement (p5, lines 111-113) and added: “Respondents were informed about the research on the questionnaire website, after which they could start the questionnaire. They were advised that filling out the questionnaire constituted their consent to participate in the study.” We uploaded the exemption letter from the Medical Ethics Committee with the revision. 

We have made the data available via Zenodo, doi 10.5281/zenodo.8195625

We included the abstract after the title page. 

We included the full ethics statement in the Methods section.

We reviewed our reference list. 

Reviewer #1:

Thank you for the opportunity to review this extremely well written mixed-methods study that contributes to the evidence base informing appropriate maternity care for women with a history of sexual violence. It is approached in a sensitive and engaging way with clear practice and research recommendations. Some minor issues to address are detailed below:

 We thank the reviewer for the positive comments and for carefully considering improvements for our paper. They were very useful. 

Abstract: The setting is not clear in the title or abstract. It would also be useful to have a very brief overview of maternity care provision in the Netherlands for context here.

Abbreviations should be written in full at first mention.

 Due to the limitation of 300 words for the abstract we can only provide a brief overview of the maternity care setting in the Netherlands. We specified that the study was carried out in The Netherlands (p2, line 32) that midwifery-led care is for low-risk pregnancy and obstetrician-led care is for high risk pregnancy (p2, line 38). 

Introduction:

Line 60: When describing Dutch prenatal care guidelines it would be useful to elaborate on the purpose for universal screening. For example is there adequate support service available if sexual violence is disclosed? Are professionals trained in how to screen for SV? Does this include any kind of trauma informed training?

 We clarified these issues (p4, lines 87-91) by adding: “The guidelines do not offer specific recommendations about the conditions, method or wording for the screening or pathways of care after a disclosure (15,16). The midwifery guideline advises extra care with intimate procedures and preparing the women for these procedures and referral for psychological help if necessary (15). The obstetric and midwifery guidelines are taught in graduate education.“ 

Methods: Although it is clear that relevant organisations and individuals were approached to disseminate the questionnaire, other public involvement and engagement is not mentioned in the methods. Were any service users with lived experience involved in the design or analysis of the research? If not it would be useful to revisit this in the discussion under 'strengths and limitations' to inform future research in this area.

 We added a description of the involvement of service users in the Methods section (p5, lines 118-120): “Two client representatives from the Dutch client organisation Birth Movement were involved in developing and piloting the questionnaire.” Furthermore, we added the following to the Discussion section (p22, lines 461-463): “Another limitation of this study is that no service users with lived experience were involved in the design or analysis of the research. Representatives of the Dutch Birth Movement were however involved in the design of the study.”

Line 107: Regarding background characteristics of participants: Using country of birth (or parents country of birth) an interesting measure of ethnicity and I think it would be useful to highlight it's limitations.Fpr example does it account for 2nd/3rd etc generation migrants, or those who were born in the Netherlands who identify as Black? This is important as those who identify as having an ethnic minority background may not be identified using 'country of birth' but still have different experiences of care such as discrimination and stigma based on their ethnicity.

 We thank the reviewer for raising this interesting point. To operationalise ethnicity, we followed the guidelines of the Dutch National Statistics Bureau. This operationalisation method comprises first- and second-generation immigrants, but not third-generation migrants. We agree that this may exclude some people who identify as Black. However, we expect this proportion to be small. In 2016 (no more recent statistics available) only 1,2% of the Dutch population was a non-Western third-generation immigrant (at least one grandparent born in a non-Western country) , of which 5 in 6 immigrants were younger than 18 years old. 

(see https://www.cbs.nl/nl-nl/nieuws/2016/47/wie-zijn-de-derde-generatie-)

We added the following to clarify this in the Methods section (p7, lines 159-163): “Consequently, Moroccan, Turkish, Surinamese, Antillean and other non-Western ethnicities in our study include first and second generation immigrants, but not third generation immigrants. The proportion of non-Western third-generation immigrants in the reproductive phase in the Netherlands is estimated to be about 1% [19]”

Line 130: Please clarify which maternity setting is the normal pathway for high/low risk women, for example do those women with 'high risk' obstetric or medical factors always experience hospital care as this is the only obstetric led setting?

 We elaborated on the organisation of care at the start of pregnancy and added (p7-8, lines 166-173); “In the Netherlands, depending on the perceived level of risk in pregnancy, clients enter maternity care and have their initial booking appointment either through a community-based midwifery practice (low risk) or through the hospital (high risk). In some regions all women have the initial booking appointment at a community-based midwifery practice and are subsequently referred to hospital when considered high risk. Respondents were grouped as having received either ‘midwife-led care’ or ‘obstetrician-led care’ at the start of pregnancy. This indicates where the booking appointment took place and thus who performed SV screening as part of history taking.” 

Line 135: What was the justification to only provide characteristics for those women who disclosed SV experience rather than compare characteristics for those who did and did not disclose? Is this something that might be presented in a further analysis of the data?

 Our study sample only includes the women who reported/disclosed an SV experience in the questionnaire (or: with a known SV experience). The research questions of this paper only concern women with a known SV experience. Indeed, in a forthcoming paper, we will investigate and report on the total sample and the differences between women with and without a known SV experience. This was, however, not the objective of the current study. 

Figure 1 was made to clarify the selection of the current study sample.

Line 157: It is not clear how the critical realist sense was incorporated into the analysis, perhaps an example would be useful here?

 We added an example to clarify (p9, lines 202-209): “For example, one respondent answered “it was only once” to the question why she had not disclosed her SV experience. To this respondent, this ‘observable’ fact evidently led to non-disclosure. A single SV event is, however, not taken literally as a reason not to disclose because, for others, a single SV event could be highly relevant to the pregnancy. The answer was understood as the respondent explaining why she felt her SV experience was not severe and therefore did not bear relevance for her in the context of maternity care. Furthermore, explaining why the experience was not so bad, was understood as a particular way to construct the SV narrative so that it was a chapter outside of the pregnancy.” 

Results: A short paragraph describing the characteristics presented in Table 1 is required. Again, is there any scope to compare these with all questionnaire respondents (not necessarily in this paper but in a future analysis to explore characteristics associated with experience of lifetime SV)?

 We added a brief description of the respondents’ characteristics (p10, lines 231-235): ”Most respondents were between 25 and 34 years of age (72,5%), had a Dutch or other Western ethnicity (92.8%), completed advanced education (vocational or university, 63,5%) and were in a relationship (97,2%). Furthermore, over half of the respondents had given birth to their first child (58,3%) and most entered maternity care through a community-based midwifery practice (81,9%) (Table 1).” 

 As described above, a forthcoming paper will address differences between women with and without a known SV experience. 

Line 333: I wondered if these quotations were better placed under theme 2 as they seem to better relate to keeping the narrative and perceptions of stigma/fear of disclosure?

 This is an interesting suggestion, indeed the underlying reason for not wanting the SV experience written down in the file is fear of stigma and could also be put with the second theme as such. However, we think it is up to the care provider to explain how sensitive information is dealt with including note taking in the file and discuss the option of not writing it down. This is not common practice in the Netherlands. Therefore we choose to keep it under the theme that care providers (can) have influence over. 

Discussion:

Line 359: Does the midwife led care model always offer continuity of care? To what extent? How many women receive continuity of care in the Netherlands? What is the policy background? It seemed to me that a lack of continuity was evident in the qualitative findings or dis this only relate to the hospital/obstetric led setting?

 Indeed, in the Netherlands, midwife-led care does not always offer continuity of care, but it often does, and more often than obstetrician-led care. We have nuanced this distinction (p19 lines 384-386,) by adding; 

“The midwife-led care model with one midwife or a small team of midwives providing prenatal, natal and postnatal care, offers continuity of care more often than an obstetrician-led care team that generally constitutes all care providers from the obstetric unit.”

Line 374: There is a substantial evidence base of black women not being listened to or believed that might be a useful contribution to this discussion. It might also be useful to highlight potential language barriers around disclosure here.

 We added the suggested and very important explanation about being believed less often as a black woman (p19, lines 402-404): 

“This delegitimisation of black women as victims leads to being believed less often after disclosure of SV (24,25). Fear of stigmatisation and not being believed may consequently inhibit an SV disclosure in black women.” 

Although we agree that a language barrier is likely to inhibit disclosure, the women in our study were proficient in Dutch, since they filled out the questionnaire. We, therefore, cannot explain this outcome by language barrier in the current study. 

Line 378: The last sentence needs further explanation to make sense.

 We added the following to introduce the last sentence better (p19, lines 407-411): “In other words, the background and obstetric characteristics we investigated were not the most important reasons why some women did, and others did not disclose.” and adjusted the last sentence to: “ Therefore, the reasons for non-disclosure that women gave in the qualitative part of our study may be more relevant for non-disclosure than the investigated respondent characteristics.”

Line 383: What is it about written information what women might fear?

We added the explanation in the sentence (p19, lines 414-415): “concerns about stigmatisation after the SV narrative was written down in the file” 

Line 423: Please expand on this point; what was the main focus of the questionnaire?

 We elaborated on the questionnaire (p22, lines 456-457) by adding: “ The main aim of the questionnaire was to evaluate women’s experiences of the care they received during their last birth, with a focus on respectful maternity care.” 

Regarding future research recommendations, it would be useful to explore the experiences of those who did disclose SV to a maternity care provider. For example how did they experience access to support services? Were fears of social care involvement relieved/substantiated and how?

 We added this recommendation (p22, lines 472-475) : “Furthermore, as our study focused on non-disclosure, future research should investigate how women who disclosed an SV experience to their maternity care provider evaluate the support they were offered after a disclosure and whether they felt stigmatised or otherwise treated negatively after a disclosure.’

---

## [Editor Report · Decision Letter 1]

19 Sep 2023

(Non-)disclosure of lifetime sexual violence in maternity care: Disclosure rate, associated characteristics and reasons for non-disclosure

PONE-D-23-12676R1

Dear Dr. de Klerk,

We’re pleased to inform you that your manuscript has been judged scientifically suitable for publication and will be formally accepted for publication once it meets all outstanding technical requirements.

Kind regards,

I. Marion Sumari-de Boer, Ph.D

Academic Editor

PLOS ONE
---

## [Editor Report · Acceptance letter]

25 Sep 2023

PONE-D-23-12676R1 

(Non-)disclosure of lifetime sexual violence in maternity care: Disclosure rate, associated characteristics and reasons for non-disclosure 

Dear Dr. de Klerk:

I'm pleased to inform you that your manuscript has been deemed suitable for publication in PLOS ONE. Congratulations! Your manuscript is now with our production department. 

Kind regards, 

on behalf of

Dr. I. Marion Sumari-de Boer 

Academic Editor

PLOS ONE